

# Multi-angle information aggregation for inductive temporal graph embedding

Shaohan Wei

School of Computing and Information Science, Fuzhou Institute of Technology, Fuzhou, Fujian, China

## ABSTRACT

Graph embedding has gained significant popularity due to its ability to represent large-scale graph data by mapping nodes to a low-dimensional space. However, most of the existing research in this field has focused on transductive learning, where fixed node embeddings are generated by training the entire graph. This approach is not well-suited for temporal graphs that undergo continuous changes with the addition of new nodes and interactions. To address this limitation, we propose an inductive temporal graph embedding method called MIAN (Multi-angle Information Aggregation Network). The key focus of MIAN is to design an aggregation function that combines multi-angle information for generating node embeddings. Specifically, we divide the information into different angles, including neighborhood, temporal, and environment. Each angle of information is modeled and mined independently, and then fed into an improved gated recuttent unit (GRU) module to effectively combine them. To assess the performance of MIAN, we conduct extensive experiments on various real-world datasets and compare its results with several state-of-the-art baseline methods across diverse tasks. The experimental findings demonstrate that MIAN outperforms these methods.

## INTRODUCTION

In the real world, various types of graph data exist, including communication graphs, citation graphs (*Jia & Yao, 2024*), brain graphs (*Ma et al., 2024*), and online consultation graphs (*Luo & Guo, 2024*). The fields of machine learning and data mining have shown significant interest in understanding and learning from such graph data (*Cui et al., 2019*; *Lin, Wang & Lin, 2024*). Graph embedding has emerged as a popular method for representing graphs by mapping nodes to a low-dimensional space (*Cao, Lu & Xu, 2015*; *Liang et al., 2024*). The node embeddings generated by graph embedding can be applied to downstream machine learning tasks, such as node classification (*Bruna et al., 2014*), link prediction (*Liu et al., 2024*), and community detection (*Wang et al., 2017*).

Traditional research in this area has primarily focused on transductive learning, which involves generating node embeddings by training the entire graph as a one-time process (*Srinivasan & Ribeiro, 2020*). However, real-world graphs undergo frequent changes, including the addition of new nodes and the occurrence of new interactions. In such cases, transductive learning would require retraining the entire graph to obtain updated node embeddings, which is not practical for real-world graphs. Assuming that

Corresponding author
Shaohan Wei,
weish_fit_edu@163.com

for a graph dataset with 10,000 nodes and 100,000 interactions, it may take 20 epochs, or about half an hour, to complete training. When a new node is added, it will take half an hour to retrain each time. In the real world, the size of this graph dataset may increase by a hundred times, and the frequency of new nodes joining may be reduced to every hour. In this case, there will be a situation where new data is added before the previous round of retraining is completed, which is unrealistic.

In contrast to transductive learning, inductive learning (*Trivedi et al., 2019*) shifts the focus from generating final node embeddings to developing models capable of dynamically generating node embeddings over time, even for previously unseen nodes. This approach facilitates flexible training and testing of models with dynamically changing real-world data, such as temporal graph data. Temporal graph is an important form in graph structured data. It changes the storage method of node interactions from adjacency matrix to interaction sequence, thus gaining the ability to dynamically increase interactions. In this data form, the inductive graph learning method is undoubtedly more suitable for temporal graphs.

Based on the aforementioned motivations, we have devised MIAN, an inductive graph embedding model. By formulating an aggregation function that combines multi-angle information for node embedding, we achieve a flexible paradigm that eliminates the need for training from scratch when incorporating node interactions. Multi-angle information is very important in the study of temporal graph embedding. In the work of *Liang et al. (2024)*, it is pointed out that time information, structural information and possible multi-modal information are all very helpful for model mining and generating higher-quality node embedding. All this information can be regarded as multi-angle information, which can expand the receptive field of the model from different aspects.

This scholarly article introduces MIAN, a method for inductive graph embedding specifically designed to learn node embeddings in temporal graphs. MIAN effectively captures graph changes, enabling the generation of node embeddings at any given time. This method places particular emphasis on the aggregation of multi-angle information to facilitate inductive temporal graph embedding.

In particular, we identify the presence of multi-angle information in the temporal graph data, which offers a more comprehensive receptive field in comparison to prior approaches that often only consider single-angle neighborhood information. This information can be categorized into different angles: (1) the neighborhood angle, which focuses on how a node's neighbor information is transmitted to the node itself through the message propagation mechanism; (2) the temporal angle, which focuses on how nodes' interactions at different moments adaptively assign varying weights for aggregation; and (3) the environment angle, which focuses on the evolution of the overall graph environment through the dynamic interaction of nodes and how this environment impacts the nodes. The information from these three angles is modeled independently and subsequently combined using an enhanced GRU module to inductively generate the corresponding node embedding for each node. This embedding naturally updates as nodes interact and evolve over time.

We conduct extensive experiments using multiple real-world datasets, comparing the performance of MIAN against several state-of-the-art baseline methods. The experimental

results demonstrate that MIAN surpasses the baseline methods, highlighting its efficiency in capturing graph changes. Our contributions can be summarized as follows:

**Problem statement:** We identify that inductive learning is better suited for temporal graphs, which dynamically and flexibly record node interactions. In addition, inductive learning can more flexibly explore multi-angle information.

**Algorithm proposal:** We propose the MIAN method, which aggregates multi-angle information in an inductive manner to dynamically generate node embeddings. We also propose an improved GRU to help the model more comprehensively aggregate multi-angle information.

**Experimental evaluation:** We empirically evaluate MIAN using various real-world datasets, showcasing its superior performance. We select datasets from different fields such as academia, business, and social network, with the number of nodes ranging from thousands to tens of thousands. The selected comparison methods are all classic methods in temporal graph learning, which have been cited many times for comparison with other works.

This paper is organized as follows: In 'Related Work', we introduce related work on temporal graph embedding and inductive learning. 'Method' presents the details of the proposed method MIAN. 'Experiment' reports the detailed settings and results of the experiments. Finally, we conclude with a conclusion.

## RELATED WORK

Graph embedding has garnered significant attention from both academic and industrial circles, finding applications in various real-world scenarios. With the increasing amount of research conducted in this field, graph embedding has progressed in multiple directions. Depending on the characteristics of the dataset, graph embedding can be classified into static graph learning and dynamic graph learning. Furthermore, considering the training patterns employed, graph embedding can be categorized into transductive learning and inductive learning.

### Temporal graph embedding

In the field of graph embedding, researchers commonly categorize graphs into two main types: static graphs and dynamic graphs. A static graph refers to a graph where neither the topological structure nor the node attributes change over time. In the early stages of graph embedding, researchers primarily focused on the topological structure of graphs. They obtained the adjacency matrix of the graphs and utilized techniques such as random walk or matrix decomposition (*Ou et al., 2016*) to learn node embeddings. Random walk means that, in a simple one-dimensional random walk, an entity starts at a specific point and at each time step, it moves either left or right with equal probability. The path taken by the entity forms a random walk. Matrix decomposition, also known as matrix factorization, is the process of breaking down a matrix into simpler, more manageable parts. This is commonly done to simplify calculations, extract meaningful information, or reduce the complexity of a problem.

For instance, DeepWalk performs a random walk procedure over the graph and then applies the Skip-Gram (*Mikolov et al., 2013a*) model to learn node embeddings (*Perozzi, Al-Rfou' & Skiena, 2014*). LINE learns node embeddings by considering first-order and second-order proximity (*Tang et al., 2015*), while node2vec introduces a biased random walk procedure to balance breadth-first and depth-first search strategies (*Grover & Leskovec, 2016*).

In contrast to static graphs, dynamic graphs encompass the changes that occur in a graph over time. Compared with static graphs, dynamic graphs can better reflect the real changes in the real world. Whether in academic citation scenarios, social scenarios, or business scenarios, the establishment of edges is never completed at the same time, and there may be a long time interval between them. If you cannot grasp this time information, you may make wrong judgments. Capturing the temporal evolution of graph structures allows researchers to obtain more effective embeddings. In the early stages of dynamic graph research, graphs were divided into several states based on timestamps, with each state representing a static snapshot of the dynamic graph (*Liang et al., 2023*). Current work often employs graph neural network (GNN) frameworks to learn node embeddings within each static snapshot at different timestamps, leveraging recurrent neural network (RNN) frameworks to capture temporal changes in node embeddings (*Xu et al., 2019*). For example, DySAT computes node embeddings through joint self-attention across structural neighborhoods and temporal dynamics (*Sankar et al., 2020*), while EvolveGCN captures dynamic graph changes by using an RNN to evolve the parameters of the graph convolutional network (GCN) (*Pareja et al., 2020*). These methods can utilize time information without changing the classic static graph technology, and also make researchers think more deeply about whether they must stick to the adjacency matrix training model. In this case, the temporal graph came into being.

Recognizing that static snapshots may not accurately represent graph changes, researchers have shifted their focus toward learning node embeddings in temporal graphs characterized by chronological interactive events (*Liu & Liu, 2021*; *Fan, Liu & Liu, 2022*). CTDNE incorporates temporal information into node embeddings through biased or unbiased random walk procedures (*Nguyen et al., 2018*), while HTNE employs the Hawkes process to capture the influence of historical neighbors on the current node to obtain node embeddings (*Zuo et al., 2018*). JODIE models the future trajectory of node embeddings and introduces a novel projection operator that learns to estimate node embeddings at any future time (*Kumar, Zhang & Leskovec, 2018*). AGLI (*Liu, Wu & Liu, 2022*) focuses on the global influence and local influence aggregation. TGC (*Liu et al., 2024b*) is the first work to focus on deep temporal graph clustering. The information comparison of these methods is shown in Table 1.

## Transductive learning and inductive learning

Graph embedding approaches can be categorized into two types based on the training pattern: transductive learning and inductive learning.

Transductive learning aims to generate fixed node embeddings by directly optimizing the final state of the graph. Many existing approaches for node embedding fall under

**Table 1  Details of different graph methods.**

| # Types | Static | Dynamic | Temporal |
|---|---|---|---|
| # Data | Adjacency matrix | Static snapshot | Adjacency list |
| # Methods | Deepwalk (Random Walk) | DySAT (RNN) | CTDNE (Random Walk) |
| | LINE (Matrix Factorization) | EvolveGCN (GCN+RNN) | HTNE (Hawkes Process) |
| | node2vec (Random Walk) | – | JODIE (RNN) |
| | – | – | AGLI (Hawkes Process) |
| | – | – | TGC (KL-Distribution) |

this category. However, transductive learning has a drawback in dynamic graphs: when the graph undergoes changes, these approaches require retraining the entire graph to obtain updated node embeddings, which can be computationally expensive. Consequently, transductive learning is not well-suited for generating new node embeddings in dynamic graphs. As mentioned above, assume that for a graph dataset with 10,000 nodes and 100,000 interactions, it may take 20 epochs, or about half an hour, to complete training. When a new node is added, it will take half an hour to retrain each time. In the real world, the size of this graph dataset may increase by a hundred times, and the frequency of new nodes joining may be reduced to every hour. In this case, there will be a situation where new data is added before the previous round of retraining is completed, which is unrealistic.

In contrast, inductive learning focuses on learning a model that can generate node embeddings at any given time. This approach does not produce fixed embeddings but instead emphasizes the ability to calculate node embeddings directly when new nodes are added. The model leverages the node's features and neighborhood information to compute its embedding. For instance, GraphSAGE learns a function that generates embeddings by sampling and aggregating features from a node's local neighborhood (*Hamilton, Ying & Leskovec, 2017*). DeepGL is a deep hierarchical framework designed for large graphs, capable of discovering both node and edge features (*Rossi, Zhou & Ahmed, 2018*). DyREP introduces a two-time scale deep temporal point process model to capture the interleaved dynamics of observed processes (*Trivedi et al., 2019*).

Based on this classification, our proposed method MIAN falls into the category of inductive learning in dynamic temporal graphs. Temporal graphs offer an accurate embedding of graph changes in real-world datasets, while inductive learning enables flexible capturing of these changes. Therefore, MIAN is better suited for generating effective node embeddings in dynamic temporal graphs. MIAN considers information fusion from multiple angles, focusing on the local neighborhood angle, time angle, and global environment angle, and fuses them together through the GRU module to generate node embeddings with richer information content. In the following sections, we will provide detailed insights into the MIAN method.

## METHOD

### Problem definition

As discussed previously, we can learn node embeddings in the graph inductively. According to the interaction between nodes, we can formally define the temporal graph as follows.

**Definition 1: Temporal graph.** *When two nodes establish an interaction within the graph, the interaction is accompanied by a clearly defined timestamp. Accordingly, the temporal graph can be formally defined as $G = (V, E, T)$, where $V$ represents the set of nodes, $E \subseteq V \times V$ denotes the set of edges, and $T$ represents the set of interaction timestamps. For each edge $e = (u, v)$ connecting node $u$ and node $v$, there exists at least one corresponding interaction instance denoted as $T_{u,v} = (u, v, t_1), (u, v, t_2), \ldots, (u, v, t_n)$, where $t_i$ represents an individual timestamp associated with the interaction between nodes u and v.*

In a temporal graph, a node engages in multiple interactions with other nodes, and these interactions can be arranged in chronological order based on their timestamps. When two nodes interact with each other, we refer to them as neighbors. The historical neighbor sequence of a node can be defined as follows.

**Definition 2: Historical neighbor sequence.**

*For a given node $u$, its historical neighbor sequence $H_u$ can be obtained, where $H_u = (v_1, t_1), (v_2, t_2), \ldots, (v_n, t_n)$. Each tuple in this sequence represents an event, indicating that node $v_i$ interacts with node $u$ at time $t_i$.*

Motivated by the work presented in *Liu, Wu & Liu (2022)*, our objective within the context of the aforementioned temporal graph is to develop a function capable of capturing multi-angle information for the purpose of inductively updating node embeddings. The following sections will provide detailed explanations of our method.

### Overall framework

As shown in Fig. 1, we consider the potential information in the temporal graph embedding from three different perspectives. Furthermore, since neighborhood-angle information and temporal-angle information are naturally close, we combine the information from these two angle to generate neighborhood with temporal information embeddings, and also generate environment information embeddings. These information embeddings are fused with the node embedding itself in the improved GRU module to generate the final node embedding.

### Neighborhood angle information

Neighborhood information plays a crucial role in graph neural networks, and many traditional graph learning methods employ message propagation mechanisms to aggregate neighborhood information for nodes. This involves combining the features of different neighboring nodes with varying weights as part of the node's own characteristics. This approach effectively enables the learning of information-rich node embeddings while maintaining the independence of each node's embedding to a certain extent.

In this process, determining how to assign weights to information from different neighboring nodes becomes an area of significant exploration. Following an interaction

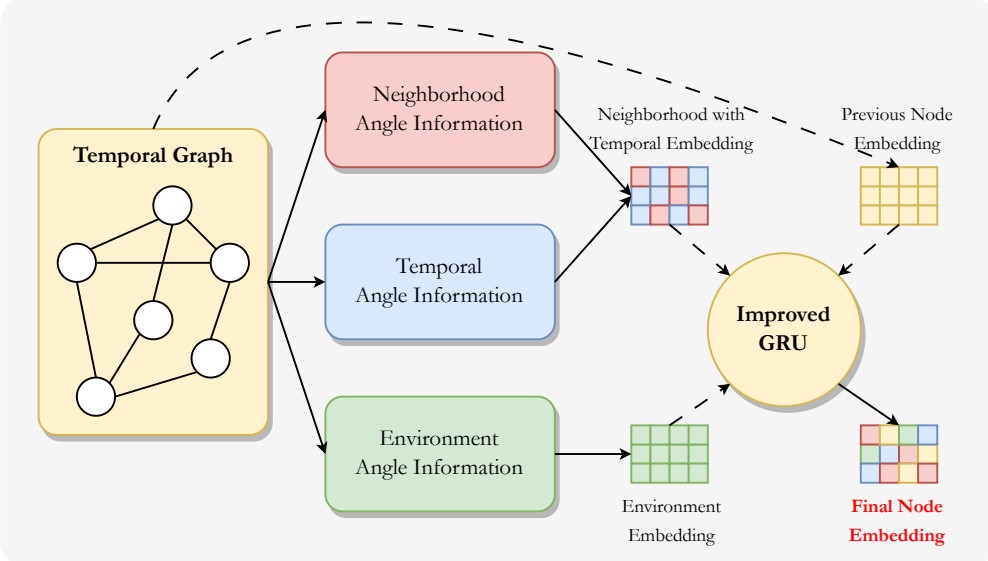

**Figure 1** **Overall framework of MIAN.** This framework consists of two steps: the first step is to model information from different angles and generate information embeddings separately, and also initialize node embeddings for later updates. The second step is to aggregate these information embeddings using an improved GRU module to generate the final (new) node embeddings.

between node $u$ and $v$, node $v$ will influence the future interactions of node $u$ with other nodes, and reciprocally, node $u$ will influence the future interactions of node $v$ with other nodes. This influence can be interpreted as different weighting methods for aggregating the characteristics of different neighboring nodes. Here, we adopt two approaches to construct such weights.

### Priority weight

The influence of a new neighbor on node $u$ diminishes as the number of neighbors increases. To illustrate this concept, we will provide an example using the movie ratings dataset, which was utilized in our experiments.

When an individual is first exposed to a particular movie genre that aligns with their taste, they are more likely to assign a high rating based on specific details. However, as they watch more movies within the same genre, their ratings for new movies in that genre may take into account various factors, such as overall quality, duration, and so on. In other words, as individuals gain experience, it becomes increasingly challenging for new movies to make a strong impression on them.

Based on the aforementioned idea, we believe that the earlier a neighbor node $i$ interacts with node $u$, the higher its priority and the greater its influence on node $u$. We calculate the priority weight $p_{u,i}$ of neighbor $i$ on node $u$ using the following approach.

$$p_{u,i} = \delta_u^p \times \frac{\exp(1/i)}{\sum_{i' \in H_u} \exp(1/i')} \tag{1}$$

Here $\delta_u^p$ is a learnable parameter that regulates the priority weight of neighbor nodes on $u$, $H_u$ is the historical neighbor sequence of node $u$, and $i$ is the sequence number of node $i$ in $H_u$. When the number of $u$'s neighbors increases, each neighbor's priority weight on node $u$ should be decreased. Thus we divide by $|H_u|$ when calculating $p_{u,i}$.

Regarding the calculation of this weight, we followed some common paradigms in classic temporal graph works. However, it should be noted that there are many ways to calculate this weight, which need to be selected and adjusted according to different model designs. For example, exponentially after calculating the position, or sharpening the weight in the form of square, or introducing an attention mechanism. These have been successfully tested in past studies.

### Affinity weight

We propose that there exists an affinity between any two nodes, which represents the closeness of their relationship. Given a node $u$ and one of its neighbor nodes $i$, we can compute their affinity $a_{u,i}$ based on their respective node embeddings $z_u$ and $z_i$ using the following approach.

$$a_{u,i} = \sigma(z_u \cdot z_i) = sigmoid(z_u \cdot z_i) \tag{2}$$

where $\sigma$ is the variant of sigmoid function to normalize its value to between $[0, 1]$, $\cdot$ denotes dot-product operation. Once we have computed the affinity between node $u$ and each of its neighbors, we can then determine the affinity weight for each neighbor using an attention mechanism. The attention mechanism is inspired by the concept that individuals tend to focus on areas of higher importance, and it has been widely utilized across various research domains.

In this context, we posit that within the historical neighbor sequence $H_u$ of node $u$, neighbors with higher affinity towards $u$ exert a greater influence on $u$. Therefore, we calculate the affinity weight $\omega_{u,i}$ for neighbor $i$ on node $u$ using the following equation.

$$\omega_{u,i} = \frac{a_{u,i}}{\sum_{i' \in H_u} a_{u,i'}}. \tag{3}$$

### Neighborhood-angle information embedding

Combining the above two weights, the embedding from neighborhood angles can be calculated as follows.

$$z_u^N = \sum_{i \in H_u} \theta_{u,i} z_i^{t_{n-1}}, \quad \theta_{u,i} = \exp(p_{u,i} \cdot \omega_{u,i}) \tag{4}$$

where $z_i^{t_{n-1}}$ is the embedding of $u$'s neighbor $i$ at time $t_{n-1}$. To calculate the influence of the next timestamp, we need to use the node embedding of the previous timestamp, which will be introduced later.

## Temporal angle information

In the realm of temporal graph learning, time information holds significant importance. Drawing inspiration from the work outlined in *Hawkes (1971)*, the Hawkes process is

employed to model discrete sequential events. This process assumes that past events have an impact on the occurrence of future events. According to this framework, the probability of future events is influenced by historical events, with the influence diminishing over time.

In the context of our problem, we posit that a node's historical neighbors play a role in shaping its future interactions. Moreover, the closer the interaction in terms of time, the greater the influence it exerts on future interactions. Consequently, we can calculate the time weight $k_{u,i}$ for neighbor $i$ on node $u$ using the following equation.

$$k_{u,i} = \exp(-\delta_u^t |t_c - t_i|) \tag{5}$$

where $t_i$ is the timestamp when neighbor $i$ interacts with $u$, $t_c$ is the current time, $\delta_u^t$ is a learnable weight parameter that regulates the neighbor nodes' time weight on $u$.

The temporal weights calculated above will be further incorporated into the embedding of the neighborhood angles as part of the weight. In this way, the temporal with neighborhood information embedding can be calcluated as follows.

$$z_u^{NT} = \sum_{i \in H_u} \theta_{u,i} z_i^{t_{n-1}}, \quad \theta_{u,i} = \exp(p_{u,i} \cdot \omega_{u,i} \cdot k_{u,i}) \tag{6}$$

Indeed, incorporating the time weight into Eq. (4) enhances the embedding of node $u$ by considering both the neighborhood weight and the temporal aspect. In a temporal graph, the time information encapsulates the interactions between a node and its neighbors. Consequently, although these two types of information are derived from different perspectives, they are inherently interconnected and can be effectively combined to improve the overall embedding of the node.

## Environment angle information

The interactions among nodes within a graph have the potential to alter the graph's structure and properties, subsequently influencing the global environment. Conversely, the global environment can also impact the interactions between nodes. As illustrated in the accompanying figure, when a node $u$ initially becomes part of a graph, it may exhibit limited sensitivity to changes occurring in the global environment. However, once the cumulative affinity of node $u$ surpasses a certain threshold, it becomes attuned to global environment changes and can readily capture the latest updates within the graph.

This mechanism highlights the notion that nodes gradually become more responsive to the evolving global environment as they accumulate stronger relationships and affinities with their neighbors. It enables nodes to adapt and incorporate the most recent changes, allowing for a more comprehensive understanding of the graph dynamics.

### *Activation threshold*

We define a node $u$ as an active node in this context. To define the active status of a node and its cumulative affinity, we draw inspiration from the linear threshold (LT) model (*Granovetter, 1978*) commonly used in the field of information propagation (*Li et al., 2018*; *Tian, Yi-Tong & Xiao-Jun, 2011*).

The LT model posits that a node can change from an inactive state to an active state if the cumulative influence from its neighbors surpasses a certain threshold. In our case, we

set $\epsilon$ as the activation threshold, which is a hyper-parameter. Additionally, we denote $\epsilon_u$ as the cumulative affinity of node $u$, representing the sum of the affinities between node $u$ and its neighbors. The calculation for the cumulative affinity $\epsilon_u$ of node $u$ is as follows.

$$\epsilon_u = \sum_{i \in H_u} \sigma(z_u \cdot z_i) = \sum_{i \in H_u} \frac{sigmoid(z_u \cdot z_i) + 1}{2}. \tag{7}$$

If, at any given time, the cumulative affinity $\epsilon_u$ of node $u$ exceeds the activation threshold $\epsilon$, node $u$ transforms into an active status. We posit that the cumulative affinity $\epsilon_u$ between node $u$ and its neighbors can offer insights into the relationship between node $u$ and the graph, to some extent. A higher cumulative affinity implies that node $u$ is more susceptible to the influence of the global environment.

Moreover, once a node enters an active status, this status is irreversible. In other words, once node $u$ starts being influenced by the global environment, that influence will persist. For instance, once a scholar gains a deep understanding of a particular field, they will always remain sensitive to the latest ideas within that field. Therefore, after node $u$ becomes active, it becomes crucial to calculate the embedding $z_u^E$ representing the global environment information for node $u$ at time $t_n$.

### Global environment embedding

To generate an environment embedding that captures the global information on each node, we create an embedding $z^E$ that aggregates all node embeddings. Initially, we sum the node embeddings obtained from the positional encoding method. The calculation for $z^E$ is as follows, and $|V|$ is the number of nodes, and $z_u^{t_0}$ is the initial node embedding of node $u$ at time $t_0$.

$$z^E = \sum_{u \in V} z_u^{t_0} / |V|. \tag{8}$$

In a temporal graph, the graph structure and features will evolve over time. Therefore, we need to update the environment embedding over time. Similarly, in the updating process, we also believe that only nodes in the active status can be "noticed" by the whole graph and thus become part of the environment embedding. In this way, only an active node $u$'s embedding is updated from $z_u^{t_{n-1}}$ to $z_u^{t_n}$, the environment embedding $z^E$ will be updated as follows:

$$z^E := \frac{|V| \times z^E - z_u^{t_{n-1}} + z_u^{t_n}}{|V|}. \tag{9}$$

### Environment-angle infromation embedding

The information pertaining to the environment received by a node is not solely dependent on the environment embedding, but also on its dynamics. Initially, we classify the dynamics of a node, denoted as $u$, into two distinct components, namely $c_u^-$ and $c_u^+$, based on its activation time $t_a$. Here, we define $c_u^-$ as the rate of change of the embedding per unit time *prior to* the activation of node $u$. On the other hand, $c_u^+$ represents the rate of change of the embedding per unit time *subsequent to* the activation of node $u$. The calculations for $c_u^-$

and $c_u^+$ are performed by considering the difference in embeddings and the time interval in the following manner:

$$c_u^- = \frac{z_u^{t_a} - z_u^{t_0}}{t_a - t_0}, \qquad c_u^+ = \frac{z_u^{t_{n-1}} - z_u^{t_a}}{t_{n-1} - t_a} \tag{10}$$

where $z_u^{t_a}$ is the embedding when node $u$ enters an active status, $z_u^{t_0}$ is $u$'s initial embedding, and $z_u^{t_{n-1}}$ is the embedding of $u$ at time $t_{n-1}$.

The influence of the environment information on a node in the graph is determined based on its relevance to the graph after becoming active. By combining Eqs. (9) and (10), we can ultimately compute the global influence embedding $z_u^E$ of node $u$ at time $t_n$ using the following formula:

$$z_u^E = \left(\frac{1}{d}\sum_{i=1}^{d}\frac{c_{u,i}^+ - c_{u,i}^-}{c_{u,i}^+}\right) \times \delta_u^g \times z^E \tag{11}$$

where $c_{u,i}^+$ is the component of $c_u^+$ in the $i$th dimension, $c_{u,i}^-$ is the component of $c_u^-$ in the $i$th dimension, and $d$ is the dimension size.

The initial term, $\left(\frac{1}{d}\sum_{i=1}^{d}\frac{c_{u,i}^+ - c_{u,i}^-}{c_{u,i}^+}\right)$, in Eq. (11), is utilized to assess the extent to which a node is influenced by the graph. It determines the proportion at which the environment embedding is integrated into the node embedding.

The disparity between the degrees of embedding change before and after activation can effectively reflect the role of global environment influence. Hence, by calculating the difference between $c_u^+$ and $c_u^-$, we can determine the degree of embedding change influenced by the graph. The percentage of this difference within $c_u^+$ indicates the depth of the global environment influence on node $u$. It is important to note that this difference is represented as a vector, with a percentage assigned to each dimension of the vector. Consequently, we average this percentage across all dimensions to obtain the final percentage.

The second term, $\delta_u^g$, in Eq. (11), represents a learnable weight parameter that regulates the global environment influence embedding of node $u$. By multiplying the weights of the first two terms with the environment embedding $z^E$, we can ultimately compute the global environment influence embedding $z_u^E$ of node $u$.

## Aggregator function

In order to consolidate the various angle information, we propose an enhanced version of the GRU module to serve as an aggregator function. The GRU module, introduced by *Cho et al. (2014)*, is capable of capturing temporal patterns in sequential data by controlling the degree of aggregation for different information and determining the proportion of historical information to retain. In particular, We extend GRU to devise an aggregator function that combines multiple-angle information embeddings with the node embeddings from the previous timestamp to generate the node embeddings for the next timestamp. The aggregator function employed is defined as follows.

$$UG_u = \sigma(W_{UG}[z_u^{t_{n-1}} \oplus z_u^{NT} \oplus z_u^E] + b_{UG}) \tag{12}$$

$$TNG_u^{t_n} = \sigma(W_{TNG}[z_u^{t_{n-1}} \oplus z_u^{NT} \oplus z_u^E] + b_{TNG}) \tag{13}$$

$$EG_u^{t_n} = \sigma(W_{EG}[z_u^{t_{n-1}} \oplus z_u^{NT} \oplus z_u^E] + b_{EG}) \tag{14}$$

$$\tilde{z}_u^{t_n} = \tanh(W_z[z_u^{t_{n-1}} \oplus (TNG_u^{t_n} \odot z_u^{NT}) \oplus (EG_u^{t_n} \odot z_u^E)] + b_z) \tag{15}$$

$$z_u^{t_n} = (1 - UG_u) \odot z_u^{t_{n-1}} + UG_u \odot \tilde{z}_u^{t_n} \tag{16}$$

Here $\sigma$ is the sigmoid function, $\oplus$ denotes concatenation operator, $\odot$ denotes element-wise multiplication. $z_u^{NT}$, $z_u^E$ and $z_u^{t_n}$ are temporal with neighborhood information embedding, environment information embedding and node $u$'s embedding at time $t_n$ respectively. $W_{UG}, W_{TNG}, W_{EG}, W_z \in \mathbb{R}^{d \times 3d}$, $b_{UG}, b_{TNG}, b_{EG}, b_z \in \mathbb{R}^d$ are learnable parameters, $UG, TNG, EG \in \mathbb{R}^d$ are called update gate, local reset gate, and global reset gate respectively.

In this study, we introduce a modification to the GRU by dividing the reset gate into two separate gates: the temporal with neighborhood reset gate ($TNG$) and the environment reset gate ($EG$). The $TNG$ gate and $EG$ gate are utilized to control the degree of information retention for the two types of embeddings, respectively. By combining the node embedding from the previous timestamp and the two reserved information embeddings, we generate a new hidden state $\tilde{z}u^{tn}$ for the next timestamp.

Furthermore, we employ the $UG$ gate to control the degree of historical information retention. Using the node embedding $z_u^{t_{n-1}}$ from the previous timestamp and the new hidden state $\tilde{z}u^{tn}$ for the next timestamp, we derive the node embedding $z_u^{t_n}$ for the next timestamp. This recursive process allows us to calculate node embeddings iteratively.

## Loss function

In order to learn effective node embeddings in a fully unsupervised setting, we employ a graph-based loss function on the node embedding $z_u^{t_n}$ and optimize it using the Adam method proposed by *Kingma & Ba (2015)*. The graph-based loss function aims to encourage similar embeddings for neighboring nodes, while ensuring that embeddings of dissimilar nodes are significantly different. To measure the similarity between two embeddings, we utilize the negative squared Euclidean distance. The loss function is defined as follows:

$$\log L = \sum_{u \in V} \sum_{v \in H_u} \left[ \log \sigma \left( -\left\| z_u^{t_n} - z_v^{t_n} \right\|^2 \right) - Q \cdot E_{v_n \sim P_n(v)} \log \sigma \left( -\left\| z_u^{t_n} - z_{v_n}^{t_n} \right\|^2 \right) \right]. \tag{17}$$

To address the significant computational burden associated with the loss function, we employ negative sampling (*Mikolov et al., 2013b*) to optimize the loss. Negative sampling involves sampling negative examples from a distribution $P_n(v)$, where $Q$ denotes the number of negative samples. In our approach, we sample negative nodes that have not appeared in the historical neighbor sequence of node $u$. This helps alleviate the computational cost while still allowing us to train the model effectively. The algorithm procedure is given in Algorithm 1.

---

**Algorithm 1** MAIN procedure

---

**Require:** Temporal Graph _G_.

**Ensure:** Node embeddings for downstream tasks.

1:  Initialize model parameters and data loading;
2:  Fetch data by batch;
3:  **repeat**
4:   **for** each _epoch_ **do**
5:    **for** each _batch_ **do**
6:     Calculate neighborhood angle information in Section 3.3;
7:     Calculate temporal angle information in Section 3.4;
8:     Calculate environment angle information in Section 3.5;;
9:     Aggregate the multi-angle information via GRU;
10:     Optimize the loss function and update the parameters;
11:    **end for**
12:   **end for**
13:  **until** Convergence

---

**Table 2   Description of datasets.**

| # Datasets | DBLP | BitCoin | ML1M | Amazon |
|---|---|---|---|---|
| # Nodes | 28,085 | 3,783 | 9,746 | 74,526 |
| # Edges | 236,894 | 24,186 | 1,100,209 | 89,689 |
| # Labels | 10 | 7 | 5 | 5 |
| # Type | Academic | Social | Business | Business |

Note that in the link prediction task, there are some other loss functions, which are briefly introduced here for reference. Binary Cross Entropy Loss (BCELoss) is a common loss function used for binary classification tasks, where the output is a probability score between 0 and 1. Hinge loss is commonly used for binary classification tasks as well, especially in support vector machines (SVMs). It penalizes predictions that are on the wrong side of the decision boundary. Bayesian Personalized Ranking Loss (BPR Loss) is often used in collaborative filtering tasks, including link prediction in recommendation systems. It optimizes the ranking of positive interactions over negative interactions.

## EXPERIMENT

### Datasets

We report the statistical information of the following real-world graph datasets in Table 2. The datasets we selected are several public datasets commonly used in the temporal graph community, which are often used by researchers for experimental analysis. These datasets cover different scenarios such as academic, social, and business, and can more comprehensively analyze the real performance and versatility of the model.

**DBLP** (*Zuo et al., 2018*): This dataset represents a co-authorship graph in the field of Computer Science. It contains over 10,000 nodes, and for our study, we extracted 10 research fields from the DBLP website.

**BitCoin** (*Kumar et al., 2016*; *Kumar et al., 2018*): The BitCoin dataset is derived from a bitcoin trading platform called Alpha. Users on the platform rate other members on a scale of -10 (total distrust) to +10 (total trust) with steps of 1. We categorized the ratings into seven distinct categories based on every three consecutive scores.

**ML1M** (*Li, Wang & McAuley, 2020*): ML1M is a widely used movie dataset employed in various machine learning tasks. Each movie in this dataset is associated with a score assigned by users, and we choose the most frequently rated score as the label for each movie. The scores range from 1 to 5, allowing us to divide the movies into five categories.

**Amazon** (*Ni, Li & McAuley, 2019*): The Amazon dataset consists of magazine subscription data. Similar to the ML1M dataset, each magazine is assigned a score based on user ratings. We select the most frequently assigned score as the label for each magazine, and the magazines are divided into five categories accordingly.

## Baselines

We compare MIAN with multiple state-of-the-art baselines:

**CTDNE** (*Nguyen et al., 2018*) gives rise to methodologies for acquiring time-respecting embeddings from networks with continuous-time dynamics.

**HTNE** (*Zuo et al., 2018*) employs the Hawkes process to capture the influence of historical neighbors on the current node, thereby obtaining node embeddings.

**JODIE** (*Kumar, Zhang & Leskovec, 2019*) represents a coupled recurrent neural network model that learns the embedding trajectories of users and items.

**TGN** (*Rossi et al., 2020*) constitutes a versatile and efficient framework for deep learning on dynamic graphs represented as sequences of timed events.

**TREND** (*Wen & Fang, 2022*) introduces a framework that incorporates both event and node dynamics, thereby enabling a more precise modeling of events within a Hawkes process-based graph neural network.

**OTGNet** (*Feng et al., 2022*) offers a comprehensive and principled learning approach for open temporal graphs, aiming to address the aforementioned challenges.

The code of our proposed method MIAN can be find in GitHub: https://github.com/doublefish-han/MIAN.

## Experimental settings

The experimental results were obtained using a desktop computer equipped with an Intel Core i7-6800K CPU, an NVIDIA GeForce RTX 3090 GPU, 64 GB of RAM, and the PyTorch deep learning framework.

All comparison methods were utilized with their default parameters, without any modifications. For our proposed MIAN method, we set the activation threshold to 0.9, the node embedding dimension size to 128, the number of training epochs to 5/10/20, and the batch size and learning rate to 128 and 0.001, respectively.

**Table 3   Node classification performance on all datasets.** We mark the optimal results in bold and the sub-optimal results in underline.

| Methods | DBLP | | BitCoin | | ML1M | | Amazon | |
|---|---|---|---|---|---|---|---|---|
| | ACC | F1 | ACC | F1 | ACC | F1 | ACC | F1 |
| CTDNE | 62.13 ± 0.94 | 61.97 ± 0.94 | 74.66 ± 0.67 | 66.06 ± 0.82 | 58.90 ± 0.93 | 54.15 ± 1.12 | 57.33 ± 0.64 | 42.12 ± 0.53 |
| HTNE | 62.86 ± 0.82 | 62.73 ± 0.78 | 76.25 ± 1.60 | 67.97 ± 1.39 | 59.73 ± 0.63 | 57.22 ± 0.79 | 57.34 ± 0.48 | 42.11 ± 0.42 |
| JODIE | 61.40 ± 0.77 | 61.07 ± 0.82 | 72.94 ± 0.65 | 67.61 ± 0.68 | 60.29 ± 0.83 | 58.33 ± 0.42 | 57.13 ± 0.31 | 41.87 ± 0.12 |
| TGN | 60.59 ± 0.84 | 62.54 ± 1.25 | 75.89 ± 0.83 | 67.83 ± 0.62 | 60.04 ± 0.35 | 57.53 ± 0.88 | 56.55 ± 0.34 | 41.59 ± 0.41 |
| TREND | 61.53 ± 1.36 | 61.78 ± 0.87 | 76.33 ± 0.69 | 67.54 ± 0.53 | 60.62 ± 0.82 | 56.49 ± 0.83 | 57.54 ± 0.43 | 41.68 ± 0.39 |
| OTGNet | 60.38 ± 1.72 | 61.83 ± 1.69 | 76.04 ± 0.88 | 67.87 ± 0.76 | 59.87 ± 0.95 | 58.02 ± 0.91 | 57.03 ± 0.64 | 42.13 ± 0.53 |
| MIAN | **63.95 ± 0.89** | **63.47 ± 0.77** | **77.67 ± 0.69** | **68.56 ± 0.44** | **62.66 ± 0.87** | **59.91 ± 0.68** | **57.85 ± 0.49** | **42.40 ± 0.32** |

We compare MIAN with other methods on several real-world datasets under the node classiciation task. We also conduct parameter sensitivity study, ablation study, and convergence analysis to verify the effectiveness of the proposed method.

## Node classification performance

In the node classification experiment, we conduct the performance comparison between the proposed MIAN algorithm and several state-of-the-art temporal graph learning methods. The results, as reported in Table 3, demonstrate that MIAN outperformed all other methods on all datasets, providing evidence for the effectiveness of our proposed method.

Note that each result was subjected to five repetitions to determine its corresponding margin of error, revealing the relative stability of our method. Furthermore, it is worth noting that the performance improvement of our method varies across different datasets. This observation suggests that the model needs to consider varying weights of information from different perspectives depending on the distribution of the data. To delve deeper into this phenomenon, we conducted additional study on parameter sensitivity and performed ablation study on the MIAN method.

## Parameter sensitivity study

In this part, we report the effectiveness of different activate threshold values on all datasets. As mentioned above, the activate threshold controls whether a node can compare the global environment information, thus further influence the final node embeddings.

Here we set the activate threshold as $0.10, 0.25, 0.5, 0.75, 0.90$, respectively. Then we training MIAN model on these different parameter settings. In this process, all other hyper-parameters are set to default values and remain unchanged. As shown in Fig. 2, we can find that when the activate threshold comes 0.90, the performance usually become the best performance. In addition, in different temporal graph datsets, the fluctuation range of the effect caused by the change of threshold is not the same.

Indeed, for varying data distributions across different datasets, it is crucial to adaptively tune the hyper-parameter values in our model to align with the specific characteristics of each dataset. Additionally, it is worth highlighting that irrespective of how the threshold value may change, the performance fluctuation of MIAN remains within a narrow range. This observation underscores the robustness of our model, as it demonstrates the model's
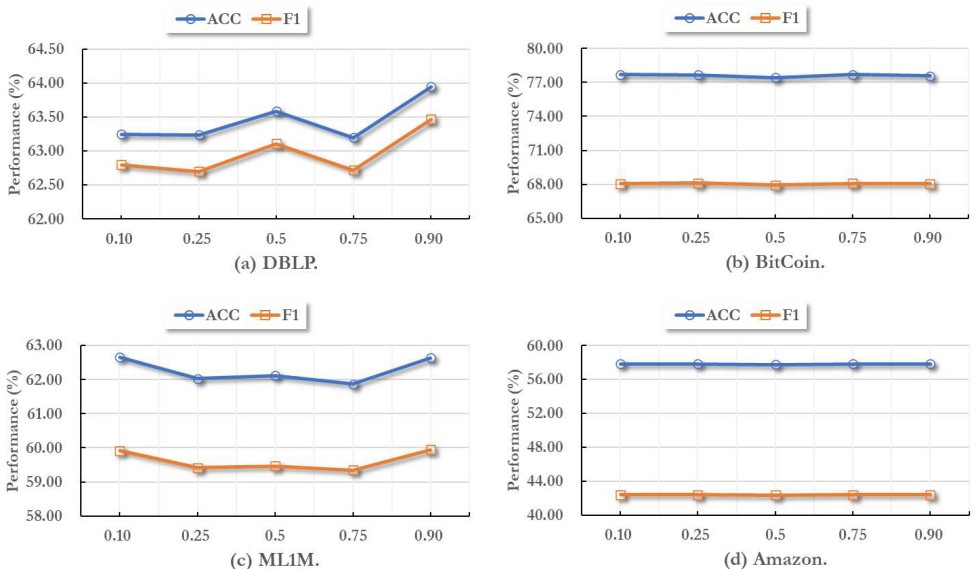

**Figure 2 Parameter sensitivity study with different node activate threshold values.**

ability to withstand variations in hyper-parameters without significant impact on its effectiveness.

Hyperparameters are values that need to be set manually. It is often difficult to find completely accurate values in practical application scenarios because the distribution and scale of real data are different, which may affect the selection of hyperparameter values. In this case, by selecting a small-scale public dataset of the same neighborhood (for example, both in the academic field) for preliminary parameter adjustment and partially extracting and adjusting parameters on internal data, researchers can often choose more reliable parameter values. The hyperparameters here are not limited to the content we discussed, but also include common hyperparameters such as embedding dimension size, training batch size, and number of epochs.

## Ablation study

We also focus on the effectiveness of different angle information on the proposed MIAN method. In this part, we conduct ablation study to observe it.

In particular, we construct several variants of MIAN. Among them, the baseline model we used is called "BASE", which does not make any changes. For information from different angles, we record them as "BASE+N", "BASE+T", and "BASE+E", respectively, and the final complete model is "MIAN".

As shown in Fig. 3, we report the ablation results on different datasets. From the figure, we can find that information from different angles improves the model performance to different extents, among which information from the neighborhood angle can bring the greatest improvement. This is consistent with our general understanding that in the field of graph embedding, information propagation brought by neighbors can greatly enhance

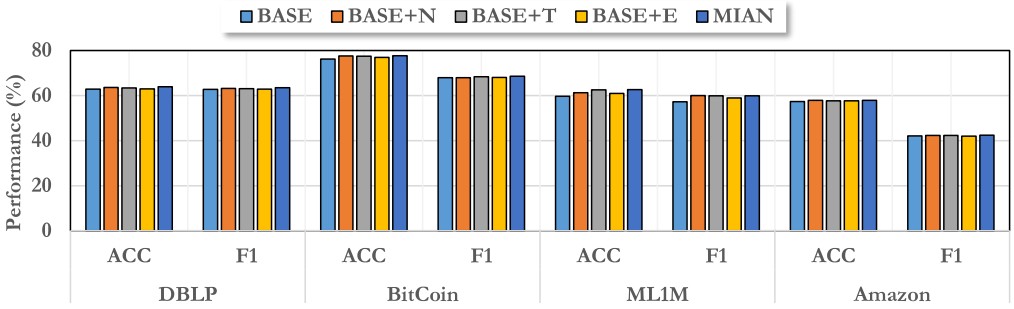

**Figure 3** Ablation study on different variants of the proposed MIAN method.

the information contained in node embedding. Regardless of the angle of information, it can improve the effect of the model, and the final complete MIAN model has the best performance. In addition, on different datasets, due to different data distributions, the improvement brought by information from different angles is different. This is also consistent with what we said above.

Note that different perspectives are of different importance, which is also well understood in real-world data. Neighborhood information is the most important, followed by time information, and then environment information. Because in static graph learning, even without time information, a good prediction rate can be achieved because we know the objects these nodes interact with. Time information is a supplement to the details. Furthermore, if the environment does not change significantly, then the impact of the global environment is not significant. If the environment fluctuates drastically, then it is worth considering the global influence to a greater extent.

## Convergence analysis

Here, we report the convergence of the model on different datasets in Fig. 4. It can be seen that our model training requires fewer epochs, that is, the loss value drops rapidly in the first five epochs, basically close to the convergence range, and then continuously adjusts in multiple epochs of training until convergence. This means that if a small loss of accuracy is acceptable, our model can be limited to a small number of training epochs, so that training can be completed quickly. Compared with other methods, the training speed is faster and the accuracy loss is lower.

In addition, we would like to point out that the change of loss function does not completely mean the change of effect. Sometimes the final adjustment of loss value does not bring about a significant improvement in performance. More specifically, the initial loss values on different datasets are different, so the thresholds of final convergence should also be different. Therefore, simply based on the change of loss function, we can only verify the speed of model convergence, but the final effect may fall on an earlier epoch. Therefore, we choose five epochs for BitCoin and ML1M, 10 epochs for DBLP, and 20 epochs for Amazon.

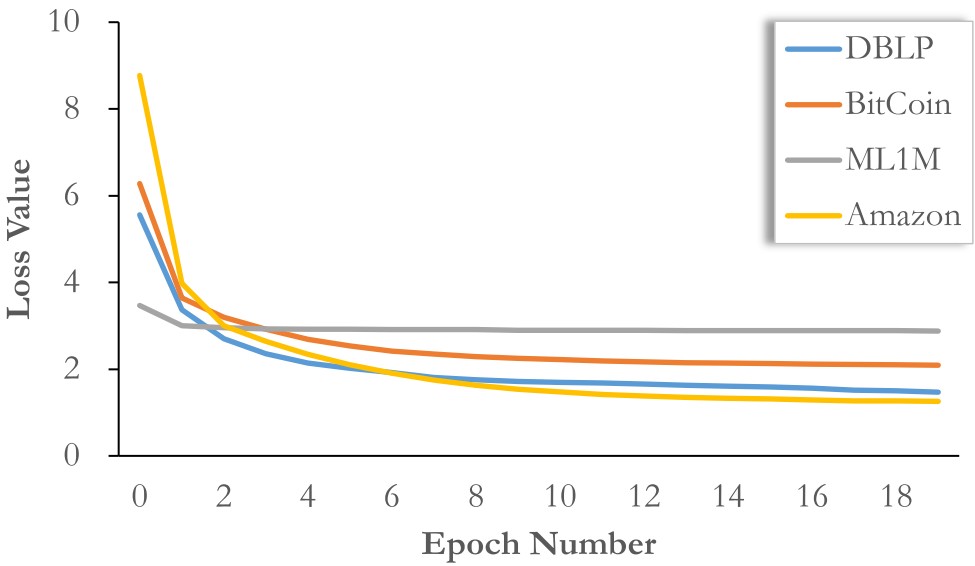

**Figure 4** Convergence analysis on all datasets with loss value evolution.

## CONCLUSION

In this paper, we propose an inductive graph embedding method MIAN that captures multiple-angle information to generate node embeddings at any time. We consider neighborhood-angle, temporal-angle, and environment-angle information separately. Then we propose an aggregator function that can flexibly capture graph changes and generate node embeddings inductively. Extensive experiments on several real-world datasets demonstrate that MIAN significantly outperforms state-of-the-art baselines. In the future, we will focus on temporal graph embedding for science application. In the real world, many data have potential connections and dynamic changes. Modeling these dynamic data as temporal graphs poses a significant challenge.

For example, in the smart city scenario, there will be information from multiple angles, including location information, interaction information, time information, and identity information. In the celestial body trajectory prediction scenario, the speed, mass, surrounding environment, and operation cycle of the celestial body are all worth considering. How to better integrate and utilize this multi-angle information will be the development direction that temporal graph learning needs to seriously consider in the future, and it is also where our proposed method really comes in handy.

### Funding
The authors received no funding for this work.

## Competing Interests

The authors declare there are no competing interests.

## Author Contributions

- Shaohan Wei conceived and designed the experiments, performed the experiments, analyzed the data, performed the computation work, prepared figures and/or tables, authored or reviewed drafts of the article, and approved the final draft.

## Data Availability

The code and data are available in the Supplemental Files.

The public datasets are available at:

- DBLP dataset: https://zuoyuan.github.io/publication/HTNE
- BitCoin dataset: https://snap.stanford.edu/data/soc-sign-bitcoin-otc.html
- ML1M dataset: https://github.com/JiachengLi1995/TiSASRec
- Amazon dataset:

https://cseweb.ucsd.edu/~jmcauley/datasets.html#amazon_reviews.

## Supplemental Information

Supplemental information for this article can be found online at http://dx.doi.org/10.7717/peerj-cs.2560#supplemental-information.

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
