# Peer review of "Multi-angle information aggregation for inductive temporal graph embedding"

_PeerJ Computer Science, doi:10.7717/peerj-cs.2560_

## Round 0.1 · original submission · Major Revisions

Dear Authors,

Please carefully address all the points raised by the 2 reviewers; do not forget to read the attached annotated manuscript by reviewer 1.

Best,

M.P

Reviewer 1 ·

Basic reporting

The revised content ensures that the article is written in clear, unambiguous, and technically correct English, adhering to professional standards of courtesy and expression. It assumes that the manuscript provides sufficient background, context, literature references, and a well-structured presentation of results and raw data.

Experimental design

The experiment shows the original primary research that fits the journal's aims and scope. It clearly defines a relevant and meaningful research question, fills an identified knowledge gap, and demonstrates rigorous investigation to high technical and ethical standards. The methodology is described with sufficient detail to allow for replication, ensuring transparency and reproducibility.

Validity of the findings

The paper communicates its impact and novelty, encourages meaningful replication, and provides all underlying data with robust statistical soundness and controlled experiments. The conclusions are well-stated, directly linked to the original research question, and limited to supporting results, making the paper a valuable and reliable contribution to the literature on temporal graph embedding.

Annotated reviews are not available for download in order to protect the identity of reviewers who chose to remain anonymous.
Cite this review as

Reviewer 2 ·

Basic reporting

All comments have been added in detail to the last section.

Experimental design

All comments have been added in detail to the last section.

Validity of the findings

All comments have been added in detail to the last section.

Additional comments

Review Report for PeerJ Computer Science
(Multi-angle information aggregation for inductive temporal graph embedding)

1. Within the scope of the study, a method called MIAN for graph embedding was proposed and compared with the state of the art methods.

2. While the introduction section explains the purpose of the study, it is recommended to add the main contributions of the study in more detail at the end of this section.

3. In the Related works section, the literature related to the study was taken from two different perspectives. However, it is recommended to add a literature table here so that the difference of the study from the literature can be clearly understood and the literature comparison can be made more clearly.

4. In the Methods section, when the framework and related steps of the proposed MIAN are examined, it is observed that it has a certain level of originality.

5. When more than one dataset is used as a dataset and the results obtained are examined, it is observed that it is at a sufficient level.

As a result, although the study is of a certain quality, attention should be paid to the above sections.

Cite this review as

---

## Round 0.2 · Major Revisions

Dear Authors,

your work has been improved by integrating the comments form the reviewers.
Nonetheless I STRONGLY recommend to put a link to a public repository (git) where you share your own code/implementation and to add in the main text a pseudo code of your algorithm.

Those are for promoting reproducibility and transparency of the results.

M.P.

Reviewer 1 ·

Basic reporting

It is written well and I don't have additional comment than the comment I have given on the first review.

Experimental design

no comment

Validity of the findings

no comment

Additional comments

n comment

Cite this review as

Reviewer 2 ·

Basic reporting

All comments have been added in detail to the last section.

Experimental design

All comments have been added in detail to the last section.

Validity of the findings

All comments have been added in detail to the last section.

Additional comments

Review Report for PeerJ Computer Science
(Multi-angle information aggregation for inductive temporal graph embedding)

Thank you for the revision. After reviewing the responses to the referee comments and the relevant changes to the paper, I recommend that the paper be accepted as the study has the potential to contribute to the literature.

Cite this review as

---

## Round 0.3 · accepted · Accept

Dear Author,

Thank you very much for accepting my suggestions, the article is now Acceptable.

M.P.